# Childhood tuberculosis outcomes and factors associated with unsuccessful treatment outcomes in selected public hospitals of Lusaka Zambia from 2015 to 2019

Dennis Ngosa[1,2]* , Joseph Lupenga[1]

1 Department of Epidemiology and Biostatistics, School of Public Health, The University of Zambia, Lusaka, Zambia, 2 Analysis Unit, Centre for Infectious Diseases Research in Zambia, Lusaka, Zambia

☯ These authors contributed equally to this work.

* dennisngosam@gmail.com

## Abstract

Treatment outcomes of tuberculosis in children are rarely evaluated. Childhood tuberculosis has been a low priority for tuberculosis programs due to difficulties in diagnosis and treatment. This study evaluated childhood tuberculosis outcomes and factors associated with unsuccessful treatment outcomes in selected public hospitals in Lusaka, Zambia from 2015 to 2019. This was a cross-sectional study conducted in eight public hospitals in Lusaka. All children aged 0–14 years, treated with tuberculosis and had treatment outcomes evaluated were included. The WHO tuberculosis treatment outcomes were grouped into successful treatment outcome (cured, treatment completed) and unsuccessful treatment outcome (death, loss to follow-up, failure). Taking unsuccessful treatment outcome as the outcome variable, logistic regression models were performed. All analyses were done at a 95% confidence interval. Out of 2,531 children managed for TB from 2015 to 2019, only 1,495 (59.1%) had treatment outcomes evaluated. Out of 1,495 participants, majority were 5 to 14 years old (50.9%), males (51.1%), HIV-negative (58.7%), and had pulmonary tuberculosis (74.2%). Bacteriological tests were performed on 59.8% of children, where 21.6% had positive bacteriological results. Bacteriologically confirmed TB was higher in children over 5 years (29.5%), pulmonary TB (25.6%), and retreatment (28.6%). The majority of children (84.2%) completed treatment, while 10.7% were cured, 1.5% were lost to follow-up, 3.1% died, and 0.5% failed treatment. Overall, unsuccessful treatment outcome was 5.1% while successful treatment outcome was 94.1%. Extrapulmonary tuberculosis was associated with unsuccessful treatment outcomes (AOR 1.64; 95% CI: 1.02–2.62). The tuberculosis successful treatment outcome met the World Health Organization's threshold goal of 90%. Children with extrapulmonary tuberculosis should be targeted as a high-risk group to improve treatment outcomes. Tracking children whose treatment outcomes were not evaluated would provide more precise estimates of TB treatment outcomes.

**Data Availability Statement:** The dataset that was analyzed for this study can be accessed using the

following link: https://figshare.com/s/9fee2b9fb8f0aed4b01c.

**Funding:** The authors received no specific funding for this work.

**Competing interests:** The authors have declared that no competing interests exist.

## Introduction

Before the coronavirus (COVID-19) pandemic, tuberculosis (TB) was the most common infectious cause of death, surpassing HIV/AIDS [1]. An estimated 10.8 million new cases of the illness and over 1.6 million estimated TB deaths were recorded globally in 2021 [1]. Out of these, 6 million were men, 3.4 million were women and 1.2 million were children [1]. In Zambia, the incidence of TB has declined significantly, from a rate of 759 per 100,000 in the year 2000 to 319 per 100,000 in 2020 [2]. In 2021, the Zambian Ministry of Health reported 50,007 cases of TB, 3,890 of which were children aged between 0–14 years, representing 7.8% of the total cases [3].

Although TB affects adults more frequently than children, children are often seen as a specific risk group. This is because TB in children can be challenging to diagnose and treat [1, 4, 5]. Furthermore, TB in children and adolescents is frequently overlooked by health providers and has been a low priority for national TB programs [6, 7].

Ending the TB epidemic by 2030 is one of the Sustainable Development Goals' health targets (6) [8]. Zambia is among the countries that have committed to the end TB by 2030 strategy, which includes implementing the 90-90-90 TB strategy (8). One of the three main goals is to achieve a 90 percent treatment success rate among those who have been identified as needing treatment [9]. However, the TB treatment outcome data (successful or unsuccessful treatment outcomes) for children 0–14 years are not reported in most countries [10]. A systematic review on treating children with TB reported that children have been left behind and neglected as data on many aspects of the treatment of childhood TB is still scarce, and the paucity of high-quality data is concerning [11].

Previous studies have shown that the outcome of TB treatment in children varies with age, sex, HIV status, and type of TB [12, 13]. Several factors put children who are diagnosed with TB disease and initiated on treatment at higher risk of experiencing unsuccessful treatment outcomes, including treatment failure, death, or being lost to follow-up. Sex, age, and type of TB have been reported to be associated with unsuccessful treatment outcomes [13, 14]. However, other studies argued that sex, age, and type of TB were not associated with unsuccessful treatment outcomes. [15].

Monitoring treatment outcomes in children with TB and understanding the factors associated with unsuccessful treatment outcomes is critical in achieving the end TB by 2030 target. In Zambia, factors associated with treatment outcomes in children have not been extensively studied. Addressing this knowledge gap provides evidence for informed decisions and the development of interventions that would help achieve a 90% success rate. This study, therefore, set out to evaluate childhood tuberculosis outcomes and factors associated with unsuccessful treatment outcomes in selected public hospitals in Lusaka, Zambia from 2015 to 2019.

## Methods

### Study design and population

The study employed a cross-sectional analysis of data from the TB registers on children aged 0 to 14 years who were diagnosed and treated for TB between 1st January 2015 and 31st December 2019. The study was conducted in the Public First-level hospitals and Third-level hospitals within the urban City of Lusaka, Zambia. This comprised six First-level hospitals (Chawama, Chelstone, Chipata, Chilenje, Kanyama, and Matero) and two Third-level hospitals (Levy Mwanawasa University Teaching Hospital (LMUTH) and University Teaching Hospital (UTH)). In Zambia, First-level hospitals are the third-largest levels of care after the Second and Third-level hospitals. They provide diagnostic, and clinical services in support of health

center referrals. On the other hand, Third-level hospitals otherwise known as University Teaching Hospitals or Tertiary hospitals are the highest referral hospitals in Zambia. All complicated cases not attended to at First or Second-level hospitals are referred to Third-level Hospitals.

Both first-level and Third-level hospitals, conduct clinical and bacteriological diagnoses of TB in children, as they are equipped with laboratory and radiological tests as well as specialized medical personnel and act as referral Hospitals for all health centers in the city of Lusaka.

**Inclusion criteria.**   The study included all children aged 0 to 14 years old, diagnosed (bacteriologically or clinically), and treated with TB from the eight public health facilities.

**Exclusion criteria.**   The study excluded children with missing and incomplete information in the TB registers.

**Sampling and sample size.**   The study sample included all children aged 0–14 years who were diagnosed and treated for TB between January 2015 and December 2019 and met the selection criteria. This gave a total of 1,495 participants.

**Variables and data extraction procedures.**   Data was extracted from TB registers from all 8 hospitals. The data extraction process was conducted from 30th November 2020 to 9th February 2021. The TB registers contained the records of all children diagnosed with TB and eligible for TB treatment, including those diagnosed with RR-TB or MDR-TB.

The data was retrieved from the TB registers using standardized extraction checklist forms that included socio-demographic characteristics (age, sex, and health facility), history of previous treatment, laboratory, and radiographic testing, HIV test results, category of TB, treatment category, and TB treatment outcomes. The diagnosis of TB was based on bacteriological and/ or clinical confirmation as described in the WHO/NTP guidelines [16].

*Definition of treatment outcomes*. Data on treatment outcomes was retrieved from the TB registers based on the standard WHO definitions of TB treatment outcomes as follows [17]; *Cured*–A pulmonary TB patient with bacteriologically confirmed TB at the beginning of treatment who was smear- or culture-negative in the last month of treatment and on at least one previous occasion; *Treatment completed*–A TB patient who completed treatment without evidence of failure BUT with no record to show that sputum smear or culture results in the last month of treatment and on at least one previous occasion were negative, either because tests were not done or because results are unavailable; *Treatment failure*–A TB patient whose sputum smear or culture is positive at month five or later during treatment; *Died*–A TB patient who dies for any reason before starting or during the course of treatment; *Lost to follow up*–A TB patient who did not start treatment or whose treatment was interrupted for two consecutive months or more; *Not evaluated*–A TB patient for whom no treatment outcome is assigned. This includes cases 'transferred out' to another treatment unit as well as cases for whom the treatment outcome is unknown to the reporting unit.

Furthermore, the final treatment outcome was dichotomized into successful treatment outcome = *cured* or *treatment completed* [17] and unsuccessful treatment outcome = *died*, *loss to follow-up*, or *treatment failure*.

## Data analysis

The data were analyzed using STATA version 17.0 software. Descriptive statistics such as frequencies and percentages were used to summarize the study variables. The proportion of bacteriologically confirmed TB was calculated by taking the number of children who had positive bacteriological results and dividing it by the number of children who underwent bacteriological TB testing. This proportion was then compared across demographic and clinical characteristics variables using the Chi-square and Fisher's exact tests.

To determine factors associated with unsuccessful treatment outcomes, logistic regression was used. The outcome variable was unsuccessful treatment outcome, a binary categorical variable which was coded = 1 if unsuccessful treatment outcome, otherwise = 0. Logistic regression was performed at adjusted and unadjusted levels. Machine-lead stepwise and investigator-lead stepwise logistic regressions were used for adjusted analysis. Robust standard errors were used when taking care of clustering by health facility. The pseudo-$R^2$ and Bayesian Information Criteria (BIC) were used to choose the best-fit model, of which the investigator lead stepwise regression was settled for, as the best-fit model. All forms of analysis in this study were done at a 95% confidence interval.

**Ethical consideration.** Ethical clearance was obtained from the University of Zambia Biomedical Research and Ethics Committee (UNZABREC) and the National Health Research Authority (NHRA) in Zambia (REF.No.1094-2020). The need to obtain informed consent from all subjects and/or their legal guardian(s) was waived by the University of Zambia Biomedical Research and Ethics Committee (UNZABREC), as the study was based on secondary data. To maintain confidentiality, names or other identifiers of study participants were not extracted from the TB registers. All methods and procedures during this study were performed in accordance with relevant guidelines and regulations (such as the Declaration of Helsinki).

## Results

### Description of the study population

A total of 2,531 children's records were extracted from the TB registers as children who were treated for TB from 2015 to 2019. Out of these, 40.9% (1036/2531) did not have their treatment outcomes evaluated as only 59.1% (1,495/2531) of the children had their treatment outcomes evaluated (Fig 1).

### Socio-demographic and clinical characteristics of participants

Overall, out of 1,495 children whose treatment outcome were evaluated, 50.9% (760/1,495) were aged 5 and 14 years, and slightly more than half were male children 51.1% (763/1,495).

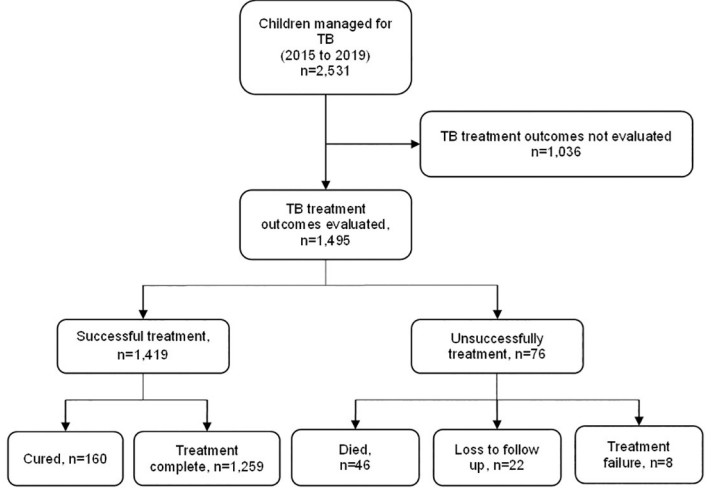

TB Tuberculosis, n Sample

**Fig 1. Study flow chart.**

**Table 1. Socio-demographic and clinical characteristics of children treated for TB (2015–2019).**

| Characteristics | Total children treated for TB N = 2,531 | | Children with evaluated treatment outcomes N = 1,495 | |
|---|---|---|---|---|
| | Frequency (n) | Percentage (%) | Frequency (n) | Percentage (%) |
| **Age** | | | | |
| 0–4 years | 1,323 | 52.3 | 732 | 49.1 |
| 5–14 years | 1,205 | 47.7 | 760 | 50.9 |
| **Sex** | | | | |
| Female | 1,196 | 47.4 | 731 | 48.9 |
| Male | 1,328 | 52.6 | 763 | 51.1 |
| **Health facility** | | | | |
| Chawama | 302 | 11.9 | 282 | 18.8 |
| Chelstone | 92 | 3.6 | 73 | 4.9 |
| Chilenje | 110 | 4.4 | 98 | 6.6 |
| Chipata | 113 | 4.5 | 101 | 6.7 |
| Kanyama | 321 | 12.7 | 299 | 20.0 |
| LMUTH | 125 | 4.9 | 31 | 2.1 |
| Matero | 553 | 21.8 | 342 | 22.9 |
| UTH | 915 | 36.1 | 269 | 18.0 |
| **Type of TB** | | | | |
| Pulmonary | 1,893 | 75 | 1107 | 74.2 |
| Extrapulmonary | 631 | 25 | 385 | 25.8 |
| **HIV Status** | | | | |
| Negative | 1,410 | 55.7 | 878 | 58.7 |
| Positive | 1,037 | 41 | 592 | 39.6 |
| Unknown | 84 | 3.3 | 25 | 1.7 |
| **Patient Type** | | | | |
| New Patient | 2,454 | 97.1 | 1442 | 96.6 |
| Retreatment | 73 | 2.9 | 51 | 3.4 |
| **Treatment Type** | | | | |
| Initial regimen with first-line drug | 2,495 | 98.6 | 1471 | 98.5 |
| Retreatment regimen with first-line drug | 21 | 0.8 | 9 | 0.6 |
| Second-line treatment drug | 14 | 0.6 | 14 | 0.9 |

UTH–University Teaching Hospital, LMUTH–Levy Mwanawasa University Teaching Hospital, TB–Tuberculosis, HIV–Human Immune Virus, N–Sample, n–Frequency, %—Percentage, Retreatment—(treatment after failure, treatment after loss to follow up, relapse and transferred in).

Matero First-level Hospital had the highest number of children 22.9% (342/1,495) followed by Kanyama First-level Hospital 20.0% (299/1,495) and Chawama first-level Hospital 18.8% (282/1,495) (Table 1).

The most common type of TB was pulmonary TB 74.2% (1,107/1,495). About 58.7% (878/1,495) of children were HIV-negative. The majority of the patients 96.6% (1,442/1,495) were new patients and the initial regimen with first-line medication was used in nearly all patients 98.5% (1,471/1,495) (Table 1).

## Diagnostic confirmation of TB among children treated for TB

Out of the total 1,495 children, bacteriological tests for TB were performed on 59.8% (894/1,495), of whom, 21.6% (193/701) had a positive test result. The proportions of children who

**Table 2. Diagnostic confirmation of TB among children treated for TB, stratified by demographic and clinical characteristics (2015–2019).**

| Characteristic | Underwent bacteriological tests | | Total | P value | All children treated for TB | | Total | P value |
|---|---|---|---|---|---|---|---|---|
| | Positive n (%) | Negative n (%) | | | Bacteriological diagnosis n (%) | Clinical diagnosis n (%) | | |
| **Overall** | 193 (21.6) | 701 (78.4) | 894 | | 193 (12.9) | 1,302 (87.1) | 1,495 | |
| **Age** | | | | | | | | |
| 0–4 years | 37 (10.1) | 329 (89.9) | 366 | <0.001[C] | 37 (5.1) | 695 (94.9) | 732 | <0.001[C] |
| 5–14 years | 155 (29.5) | 370 (70.5) | 525 | | 155 (20.4) | 605 (79.6) | 760 | |
| **Sex** | | | | | | | | |
| Female | 103 (22.5) | 354 (77.5) | 457 | 0.491[C] | 103 (14.1) | 628 (85.9) | 731 | 0.186[C] |
| Male | 90 (20.6) | 346 (79.4) | 436 | | 90 (11.8) | 673 (88.2) | 763 | |
| **Infection Site** | | | | | | | | |
| Pulmonary | 185 (25.6) | 539 (74.4) | 724 | <0.001[C] | 185 (16.7) | 922 (83.3) | 1,107 | <0.001[C] |
| Extrapulmonary | 8 (4.8) | 159 (95.2) | 167 | | 8 (2.1) | 377 (97.9) | 385 | |
| **Patient Type** | | | | | | | | |
| New | 183 (21.3) | 676 (78.7) | 859 | 0.299[C] | 183 (12.7) | 1,259 (87.3) | 1,442 | 0.148[C] |
| Relapse | 10 (28.6) | 25 (71.4) | 35 | | 10 (19.6) | 41 (80.4) | 51 | |
| **HIV Status** | | | | | | | | |
| Negative | 119 (23.1) | 396 (76.9) | 515 | 0.424[E] | 119 (13.6) | 759 (86.5) | 878 | 0.632[E] |
| Positive | 72 (19.5) | 298 (80.5) | 370 | | 72 (12.2) | 520 (87.8) | 592 | |
| Unknown | 2 (22.2) | 7 (77.8) | 9 | | 2 (3.2) | 23 (92.0) | 25 | |

C—Chi-square test, E—Fishers exact, TB Tuberculosis, HIV Human Immune Virus, % Percentage, n Frequency

had positive test results for TB varied significantly between children under 5 years of age, 10.1% (37/366), and children aged 5 to 14 years, 29.5% (155/525).

There was a statistically significantly higher proportion of children with pulmonary TB 25.6% (185/724) who had a positive bacteriological test result compared to children with extrapulmonary TB 4.8% (8/167). The proportion of children who had a positive bacteriological test result for TB was significantly higher in retreatment patients 28.6% (10/35) compared to new patients 21.3% (183/859) but this was not statistically significant. The proportion of children who had a positive bacteriological test result did not differ significantly by HIV status or sex of the participants (p> 0.05).

Overall, only 12.9% (193/1,495) of children received treatment following a bacteriological confirmation of TB, the remaining 87.1% (1,302/1,495) received treatment based on clinical diagnosis. Children who were treated for TB clinically comprised of children who had negative bacteriological results and those who did not undergo bacteriological testing (Table 2).

## Treatment outcomes of children diagnosed and managed for TB

The TB treatment outcomes were categorized into five groups (cured, treatment complete, treatment failure, loss to follow-up, and died). The proportion of Children who had a cured treatment outcome was lowest 7.9% (33/420) in 2018 and highest 20% (34/170) in 2016. The proportion of patients who completed treatment without evidence of failure or cure was highest 88.4% (290/328) in 2017 and lowest 72.9% (124/170) in 2016. The highest proportion of deaths 4.7% (8/170) was observed in 2016 and the lowest 1.6% (3/183) was observed in 2015. On the other hand, treatment failure among children diagnosed with TB was very low throughout the five years and varied from 0 in 2015 to 1.8% (3/170) in 2016. Loss to follow-up

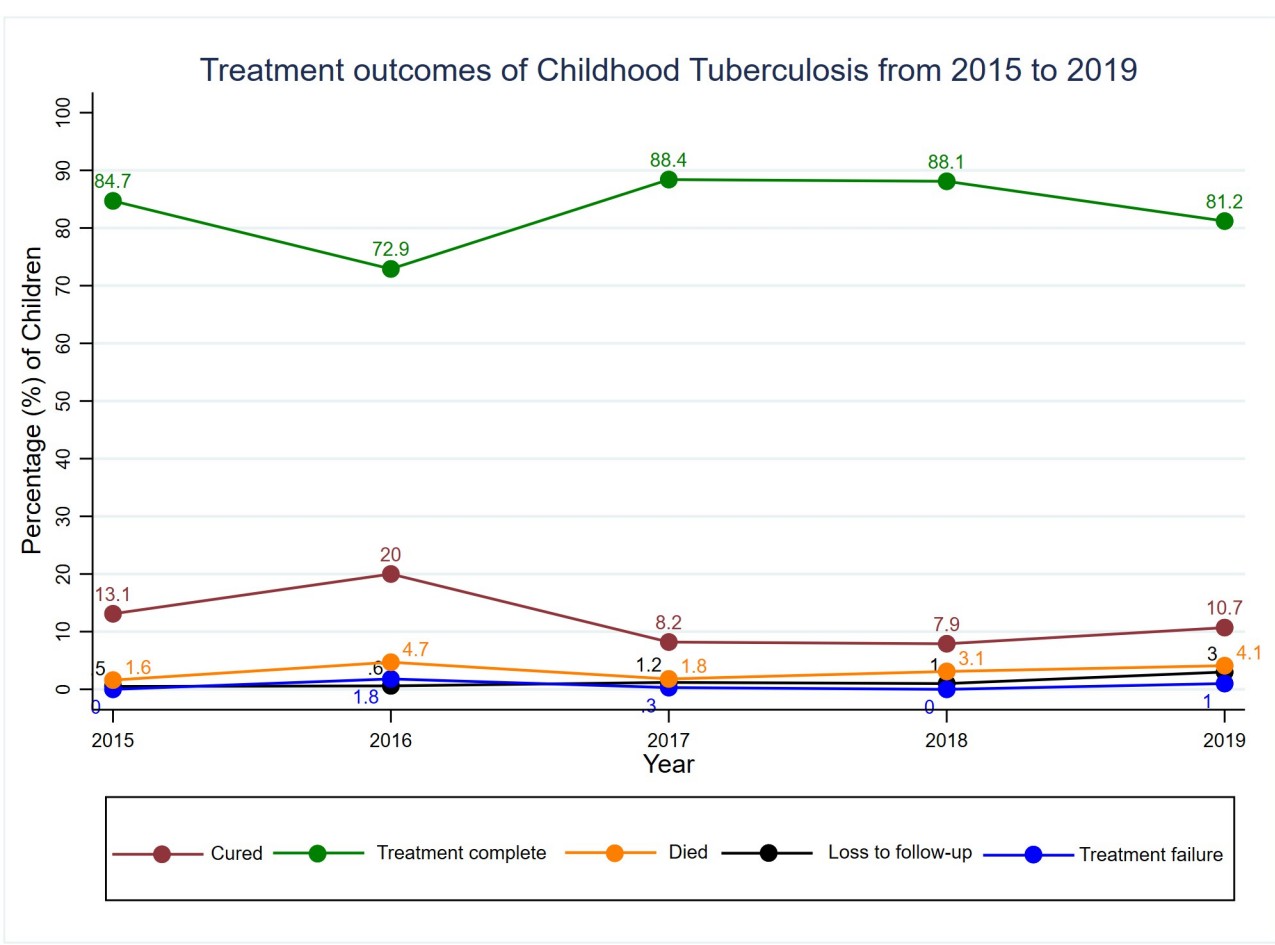

**Fig 2. Treatment outcomes of childhood tuberculosis from 2015 to 2019.**

among children on TB treatment also remained very low during the five years; the highest proportion of 3% (12/394) was recorded in 2019 (Fig 2).

The treatment outcomes were further categorized into successful (cured + treatment complete) and unsuccessful (died + loss to follow up + treatment failure) treatment outcomes. The highest proportion of unsuccessful treatment outcomes 8.1% (32/394) was recorded in 2019 and the lowest 2.2% (4/183) was recorded in 2015 (Fig 3).

The overall treatment outcome showed that, out of the total cases of Children who had their TB treatment outcome evaluated from 2015 to 2019, 10.7% (160/1,495) were cured, 84.2% (1,259/1,495) had completed treatment, 3.1% (46/1,495) died, 1.5% (22/1,495) were lost to follow up, and the remaining 0.5% (8/1,495) had failed treatment. Over the 5 years, 5.1% (76/1,495) clients had unsuccessful treatment outcomes while 94.9% (1,419/1,495) clients had successful treatment outcomes. Seven percent (27/384) of children who had extrapulmonary TB had an unsuccessful treatment outcome while only 4.4% (49/1,107) of children with pulmonary TB had an unsuccessful treatment outcome (Table 3).

**Factors associated with unsuccessful treatment outcome in children <15 years.** From the univariate analysis, age, sex, type of TB, HIV status, treatment history, and diagnostic type were all not significantly associated with the unsuccessful treatment. In multivariable logistic regression, children with extrapulmonary TB had 1.64 times greater odds of having

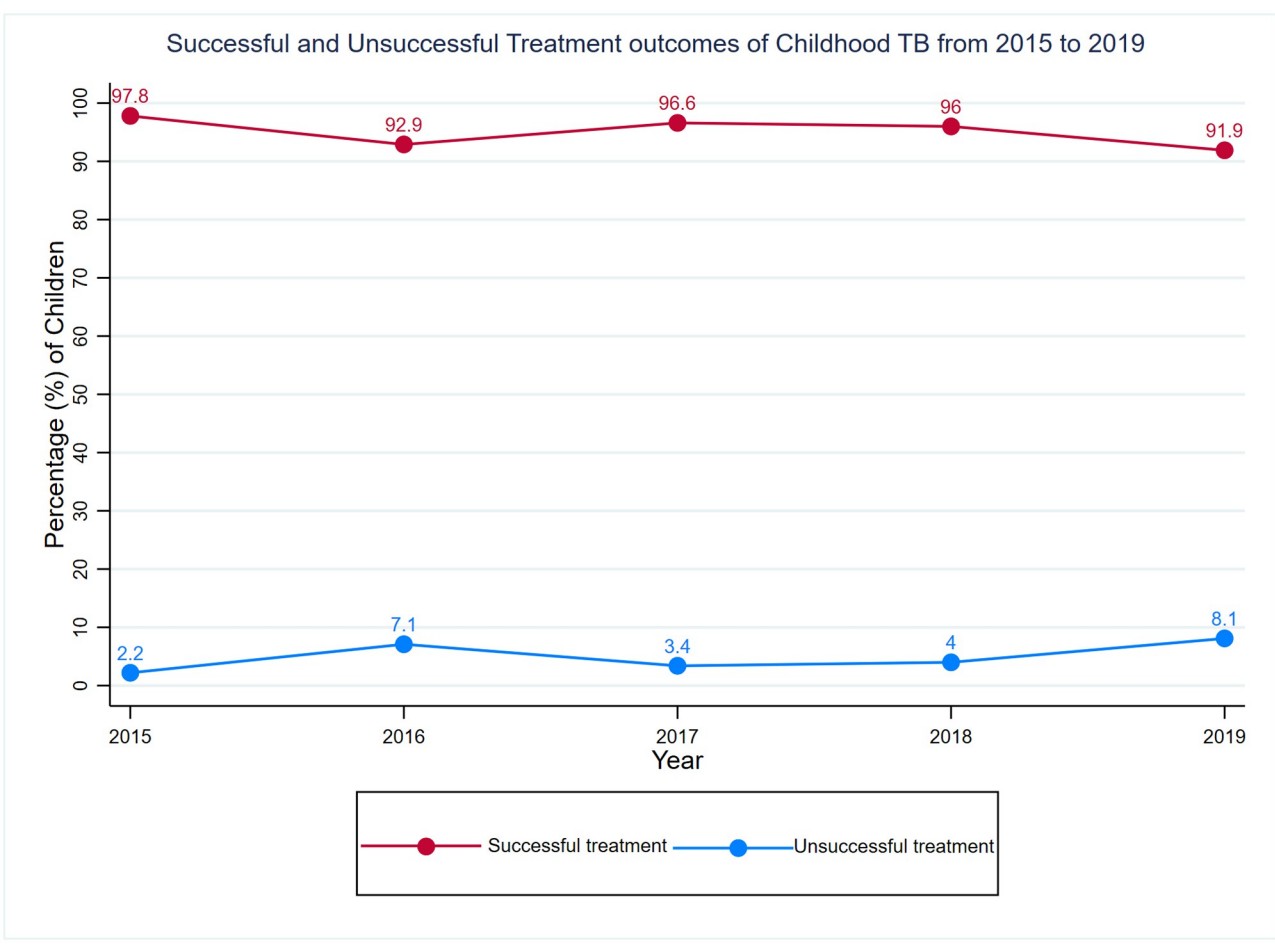

**Fig 3. Successful and unsuccessful treatment outcomes of childhood TB from 2015 to 2019.**

unsuccessful treatment outcomes than children with pulmonary TB (AOR 1.64; 95% CI: 1.02–2.62) and this was statistically significant (Table 4).

## Discussion

The current study provided valuable insights into childhood TB outcomes and factors associated with unsuccessful treatment outcome in selected public hospitals in Lusaka, Zambia from 2015 to 2019. The findings revealed that a high proportion of children who were treated for TB did not have their treatment outcomes evaluated. Among children whose treatment outcome were evaluated, majority were above the age of five, males, and were HIV-negative. Pulmonary TB was the most common type of TB. Children aged 5 to 14 years, pulmonary TB, and relapse patients had a significantly higher proportion of bacteriologically confirmed tuberculosis. The successful TB treatment outcome fell above the WHO recommended threshold of 90, as the proportion of unsuccessful treatment outcome was relatively low. Extrapulmonary TB was significantly associated with unsuccessful treatment outcomes in children.

Pulmonary TB (74.2%) was found to be the most common type of TB in children, which is consistent with earlier studies [18, 19]. Extrapulmonary TB accounted for roughly a quarter of all children who had TB, which is consistent with findings from Zimbabwe [20], Ghana [21],

**Table 3. Treatment outcomes of childhood tuberculosis (2015–2019).**

| Variable | Category of treatment outcome n (%) | | | | | | Overall treatment outcome n (%) | | |
|---|---|---|---|---|---|---|---|---|---|
| | **Cured** | **T C** | **Died** | **LTFU** | **TF** | **Total** | **ST** | **UT** | **Total** |
| **Overall total** | 160 (10.7) | 1259 (84.2) | 46 (3.1) | 22 (1.5) | 8 (0.5) | 1,495 | 1419 (94.9) | 76 (5.1) | 1,495 |
| **Health Facility** | | | | | | | | | |
| *First-level hospitals* | | | | | | | | | |
| Chawama | 36 (12.8) | 227 (80.5) | 10 (3.5) | 5 (1.8) | 4 (1.4) | 282 | 263 (93.1) | 19 (6.7) | 282 |
| Chelstone | 15 (20.5) | 47 (64.4) | 4 (5.5) | 6 (8.2) | 1 (1.4) | 73 | 62 (84.9) | 11 (15.1) | 73 |
| Chilenje | 8 (8.2) | 86 (87.8) | 4 (4.1) | 0 | 0 | 98 | 94 (95.9) | 4 (4.1) | 98 |
| Chipata | 32 (31.7) | 63 (62.4) | 5 (5.0) | 0 (0.0) | 1 (1.0) | 101 | 95 (94.1) | 6 (5.9) | 101 |
| Kanyama | 51 (17.1) | 235 (78.6) | 9 (3.0) | 2 (0.7) | 2 (0.7) | 299 | 286 (95.7) | 13 (4.3) | 299 |
| Matero | 15 (4.4) | 316 (92.4) | 5 (1.5) | 6 (1.8) | 0 | 342 | 331 (96.8) | 11 (3.2) | 342 |
| *Third-level hospitals* | | | | | | | | | |
| LMUTH | 3 (9.7) | 21 (67.7) | 5 (16.1) | 2 (6.5) | 0 | 31 | 24 (77.4) | 7 (22.6) | 31 |
| UTH | 0 | 264 (98.1) | 4 (1.5) | 1 (0.4) | 0 | 269 | 264 (98.1) | 5 (1.9) | 269 |
| **Type of TB** | | | | | | | | | |
| Extra Pulmonary | 19 (4.9) | 339 (88.1) | 14 (3.6) | 9 (2.3) | 4 (1.0) | 385 | 358 (93.0) | 27 (7.0) | 385 |
| Pulmonary | 139 (12.6) | 919 (83.0) | 32 (2.9) | 13 (1.2) | 4 (0.4) | 1107 | 1,058 (95.6) | 49 (4.4) | 1,107 |

TC—Treatment completed, LTFU—Loss to follow up, TF—Treatment failure, ST—Successful treatment, UT—Unsuccessful treatment, n—frequency, %—percentage, LMUTH—Levy Mwanawasa University Teaching Hospital, UTH—University Teaching Hospital, TB–Tuberculosis, Successful treatment- (cured + treatment complete), Unsuccessful treatment- (died + loss to follow up + treatment failure)

and Pakistan [22]. Other studies, however, found a higher proportion (33% to 56%) of extra-pulmonary TB [18, 23–25].

The proportion of bacteriologically confirmed TB among children who underwent bacteriological testing was 21.6%. This is substantially lower compared to findings from a Brazilian study (69%) [19]. On the other hand, the proportion of children who were managed for TB following bacteriological diagnosis among all children (underwent microbiological tests or not) was equally low (12.9%). These findings are consistent with previous studies elsewhere [24]. However, another study done in Nigeria showed a higher proportion of bacteriologically confirmed TB among all children who children being treated for TB (28%) [25].

The low proportion of bacteriologically confirmed TB in the current study might be attributed to the fact that bacteriological testing for TB was only performed on 59.8% of children out of the total children who were treated for TB from 2015 to 2019 and had their outcomes evaluated. This meant that the majority (87.1%) of children were treated for TB based on clinical diagnosis which was a combination of children who had negative bacteriological test results and those who did not undergo bacteriological testing but might have been subjected to other means of diagnosis such as radiographs.

During the five years (2015–2019), the death outcome in children with tuberculosis was low, ranging from 1.6% to 4.7%, which was consistent with other studies [18, 26]. One of the primary problems in estimating the proportion of children who died from TB was that the exact causes of death were not recorded in the TB registers. As a result, the observed rates may be exaggerated if some deaths were caused by factors unrelated to TB treatment.

Treatment failure among children with TB was uncommon over the five years, and this finding was within the range (0% - 1.8%) which was similar to what has been reported in other studies [24]. Furthermore, the proportion of children who were lost to follow-up on TB

**Table 4. Socio-demographic and clinical factors associated with unsuccessful treatment outcome.**

| Variable | Unsuccessful outcome n (%) N = 76 | Unadjusted | | Adjusted | |
|---|---|---|---|---|---|
| | | OR (95% CI) | P-value | OR (95% CI) | P-value |
| **Age group** | | | | | |
| 0–4 years | 43 (5.9) | 1 | | 1 | |
| 5–9 years | 19 (4.7) | 0.79 (0.58, 1.08) | 0.139 | 0.74 (0.52, 1.07) | 0.109 |
| 5–14 years | 14 (3.9) | 0.65 (0.33, 1.29) | 0.223 | 0.62 (0.32, 1.20) | 0.157 |
| **Sex** | | | | | |
| Female | 32 (4.8) | 1 | | 1 | |
| Male | 44 (5.8) | 1.34 (0.84, 2.12) | 0.217 | 1.34 (0.82, 2.18) | 0.238 |
| **Types of TB** | | | | | |
| Pulmonary | 49 (4.4) | 1 | | 1 | |
| Extra Pulmonary | 27 (7.0) | 1.63 (0.97, 2.73) | 0.063 | 1.64 (1.02, 2.62) | 0.042* |
| **HIV Status** | | | | | |
| Negative | 40 (4.6) | 1 | | 1 | |
| Positive | 34 (5.7) | 1.28 (0.62, 2.65) | 0.512 | 1.37 (0.67, 2.78) | 0.382 |
| Unknown | 2 (8.0) | 1.82 (0.43, 7.73) | 0.416 | 1.91 (0.44, 8.22) | 0.382 |
| **Treatment History** | | | | | |
| New Patient | 73 (5.1) | 1 | | | |
| Retreatment | 3 (5.9) | 1.17 (0.36, 3.76) | 0.790 | - | - |
| **Diagnostic Type** | | | | | |
| Clinical | 66 (5.1) | 1 | | 1 | |
| Bacteriological | 10 (5.2) | 1.02 (0.42, 2.51) | 0.960 | 1.43 (0.52, 3.90) | 0.484 |

TB Tuberculosis, HIV Human immune virus, CI Confidence interval,

* Statistically significant at 5% significance level, OR Odds ratio, % Percentage, n Frequency, N Sample

treatment also remained very low (0.5% - 3%). This was lower than what was reported in other studies in Ethiopia [18, 24].

A high proportion of children with a not-evaluated treatment outcome (40.9%) was observed over the five years. In contrast to the results of this study, other studies reported a much lower rate of not-evaluated treatment outcomes (0.1% - 22.5%) [13, 24, 27].

One of the potential reasons for having a high proportion of cases with not-evaluated treatment outcomes could have been transfers that happen between higher and lower-level health facilities. In the continuum of care, patients are diagnosed from higher-level health facilities that have the necessary equipment and human resources but later get transferred to lower-level health facilities that are near the patients' homes for continuation of treatment. This has been demonstrated in literature in a study that was done in Ethiopia, which found that most children with not-evaluated treatment outcomes were transferred [13]. This was also evident in the current study as the results reviewed that over 70% of children from each of the two third-level hospitals (LMUTH and UTH) (Highest level facilities in Zambia) had not-evaluated treatment outcomes as compared to the six-level First-level hospitals where the proportion of children with not-evaluated treatment outcomes was 6% to 20% with one facility reporting 38% (S1 Table).

This referral system between higher and lower-level health facilities often results in a break in follow-up, making it difficult to track the progress and the success of treatment outcomes in children. This also raises questions about the guarantee of continuity of care for these children.

Additionally, apart from transfers, poor documentation, and poor follow-up systems could have also contributed to the high proportion of children whose treatment outcomes were not evaluated. Without proper evaluation of treatment outcomes, it becomes difficult to determine the effectiveness and success rates of the treatments being provided.

The unsuccessful treatment outcome ranged between 2.2% to 8.1% (2015 to 2019) with a pooled estimate of 5.1%, and the successful treatment outcome ranged between 91.9% to 97.8% (2015 to 2019) with a pooled estimate of 94.9%. This showed that the successful treatment rate met the WHO-mandated 90% treatment success threshold. Our results were comparable to studies that reported similar successful treatment outcomes of 94.2% (Pakistan) [22], and 93.1% (Botswana) [28]. However, other studies have reported lower successful treatment outcomes 88.6% (Ethiopia) [29], 85.4% (Pakistan) [27], 83% (Nigeria) [25], and 58.4% (Nigeria) [30].

Regression analysis revealed that extrapulmonary TB had a higher risk of unsuccessful treatment outcomes. Other studies have come to the same conclusion [24, 31]. The study found no association between age, sex, HIV status, treatment history, diagnostic type, and unsuccessful treatment outcome for TB. Similar results have been reported in the past by some studies [22]. But other studies reported that age [12, 13], sex [27], HIV status [13, 32, 33], treatment history, and diagnostic method [27] were associated with unsuccessful treatment outcomes in children with TB. The inconclusiveness of our findings could again be because the sample size was reduced at regression analysis as the not-evaluated children could not be exclusively classified as successful or unsuccessful outcomes to reduce bias in findings of factors associated with unsuccessful treatment outcomes, hence were excluded from the regression analysis.

The strengths that were in this study include the fact that the sample included all children treated for TB between 2015 and 2019 who met the study's selection criteria, which gave the study a big sample size. However, some limitations were also noted; the study only focused on TB cases in children from 8 hospitals within the city of Lusaka, Zambia. Therefore, the results presented here are not representative of all the TB cases in children in Zambia. The large number of not-evaluated children that was observed could have led to overestimating or underestimating treatment outcomes. This is because not-evaluated cases were dropped from the analysis which reduced our sample from 2,531 to 1,495 as they could not be classified as successful or unsuccessful treatment outcomes. Documentation in the TB registers was inadequate, there was missing data (challenges of keeping records in health facilities). This made it difficult to establish true reasons behind children with not-evaluated treatment outcomes. Although a high number of children who had their treatment outcomes not evaluated were dropped from the study, this did not affect the distribution of clinical characteristics (Table 1).

Furthermore, there was a limitation in the number of clinical and social-demographic factors that could be investigated at regression analysis as not so many variables are captured in TB registers. There was also missing information on the details of the treatment regimen that had been given to the children and whether there was a shift or not. Few children had extrapulmonary TB as compared to those who had pulmonary TB among those who underwent microbiological testing for TB, the vast difference could have led to an overestimation of the statistical significance of positive bacteriological TB between the two groups.

## Conclusion

Among children who had treatment outcomes evaluated, pulmonary TB was the most common type of TB in children. The majority of children who were treated for TB based on clinical diagnosis. Bacteriologically confirmed TB among children who were tested for TB was low.

Overall, the successful treatment outcome was above the WHO's goal of 90% for the period 2015–2019. Children under the age of five should be targeted as a high-risk group in strategies to eliminate childhood TB/improve TB treatment success rates with a primary focus on children with extrapulmonary TB. Future studies should explore the possibility of tracking the transferred (not-evaluated) children for better estimations of TB treatment outcomes as Third-level hospitals had a very high number of children whose outcomes were not evaluated.

## Supporting information

**S1 Table. Evaluation status of treatment outcome for children treated for TB by demographic and clinical characteristics (2015–2019).**
(DOCX)

## Acknowledgments

The authors are grateful to The University of Zambia, School of Public Health, Department of Epidemiology and Biostatistics. We are also grateful to the Ministry of Health, Health facility management from all 8 facilities where data was collected and all health workers operating from the TB sections in the 8 health facilities that were included in this study.

## Author Contributions

**Conceptualization:** Dennis Ngosa, Joseph Lupenga.

**Data curation:** Dennis Ngosa, Joseph Lupenga.

**Formal analysis:** Dennis Ngosa, Joseph Lupenga.

**Investigation:** Dennis Ngosa, Joseph Lupenga.

**Methodology:** Dennis Ngosa, Joseph Lupenga.

**Project administration:** Dennis Ngosa, Joseph Lupenga.

**Resources:** Dennis Ngosa.

**Software:** Dennis Ngosa, Joseph Lupenga.

**Supervision:** Dennis Ngosa, Joseph Lupenga.

**Validation:** Dennis Ngosa, Joseph Lupenga.

**Visualization:** Dennis Ngosa, Joseph Lupenga.

**Writing – original draft:** Dennis Ngosa, Joseph Lupenga.

**Writing – review & editing:** Dennis Ngosa, Joseph Lupenga.

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
