## [Decision Letter · Decision Letter 0]

5 Sep 2023

PGPH-D-23-01445

Childhood Tuberculosis Outcomes and Factors Associated with Unsuccessful Treatment Outcomes in Selected Public Hospitals of Lusaka Zambia from 2015 to 2019

Dear Dr. Ngosa,

Thank you for submitting your manuscript to PLOS Global Public Health. After careful consideration, we feel that it has merit but does not fully meet PLOS Global Public Health’s publication criteria as it currently stands. Therefore, we invite you to submit a revised version of the manuscript that addresses the points raised during the review process.

We look forward to receiving your revised manuscript.

Kind regards,

Sizulu Moyo, MBCBH, MPH, PhD

Academic Editor

Journal Requirements:

Additional Editor Comments (if provided):

Please explain the rationale for selecting the hospitals included and where they are situated, urban, rural- could outcomes differ by this geolocation urban vs rural or by hospital? This should be investigated.

Wasn't these more information about these children? other comorbid conditions other than TB

Line 347 can the authors explain this ....diagnosed clinically, making it a challenge to evaluate the true magnitude of TB disease in children before investigating the treatment outcomes..... , since it is known that TB in young children in is paucibacillary and will have to be diagnosed clinically, and that sputum sample collection is challenge in children

Who are those not evaluated? Are they the same as the rest of the sample analyses. This a major finding that may children have no outcomes- is this across all area/ hospitals??

Reviewers' comments:

Reviewer's Responses to Questions

**Comments to the Author**

1. Does this manuscript meet PLOS Global Public Health’s publication criteria? Is the manuscript technically sound, and do the data support the conclusions? The manuscript must describe methodologically and ethically rigorous research with conclusions that are appropriately drawn based on the data presented.

Reviewer #1: Partly

Reviewer #2: Yes

2. Has the statistical analysis been performed appropriately and rigorously?

Reviewer #1: I don't know

Reviewer #2: Yes

3. Have the authors made all data underlying the findings in their manuscript fully available (please refer to the Data Availability Statement at the start of the manuscript PDF file)?

Reviewer #1: Yes

Reviewer #2: Yes

4. Is the manuscript presented in an intelligible fashion and written in standard English?

Reviewer #1: Yes

Reviewer #2: Yes

5. Review Comments to the Author

Reviewer #1: Childhood Tuberculosis Outcomes and Factors Associated with Unsuccessful Treatment Outcomes in Selected Public Hospitals of Lusaka Zambia from 2015 to 2019

PGPH-D-23-01445

Thank you for asking me to review the manuscript entitled ‘Childhood Tuberculosis Outcomes and Factors Associated with Unsuccessful Treatment Outcomes in Selected Public Hospitals of Lusaka Zambia from 2015 to 2019.’ In this manuscript, the authors report treatment outcomes and factors associated with successful versus unsuccessful outcomes in children and young adolescents 0 – 14 years of age.

I have three major concerns with this manuscript:

1. Although the authors are reporting treatment outcomes and factors associated with these outcomes, the analysis and discussion is limited and no new information or insights are presented. I am aware that this is probably due to the limited data captured in the TB registers, but wondered if any additional information could be presented. For example, was the type of extrapulmonary TB reported?

2. In lines 251 – 256, table 3, and the conclusion the authors state that over 90% of the children had a successful treatment outcome. This is not exactly correct and overstating things, as it is only if the 41% of the children ‘not evaluated’ are excluded that the successful treatment outcome rate is over 90%. This needs to be clarified in the abstract, manuscript and the table.

3. As mentioned in point 2 above, the pooled proportion of children for whom no treatment outcome was assigned is 41%. In other words, a little less than half of the children had no treatment outcome assigned. This is a very large proportion of children not evaluated and must be mentioned in the abstract and the results and not just in the third last paragraph of the discussion. In the paragraph in which this is discussed the authors suggest that this large number could be as patients are transferred to another treatment facility. However, there are other reasons as to why no treatment outcome is assigned, mostly because of lost contact with patients. These would include treatment interruption, sub-optimal adherence and even death. Are there any other journal articles which discuss reasons for ‘not evaluated’? Given the extent of the children in the not evaluated' category, this is something the authors could include in the discussion. That is, the limitations of this indicator.

Additional comments:

Abstract:

1. ‘A total enumeration of children aged 0-14 years who were diagnosed and treated for tuberculosis between 2015 and 2019 was done.’

The sentence above does not make sense. Do you mean ‘All children 0 – 14 years who were diagnosed and treated for TB between 2015 and 2019 were included’?

2. ‘Overall, (6.3%) of patients were cured, (49.7%) completed treatment, (0.9%) lost to follow-up, (1.8%) died, 0.3% failed treatment and 40.9% were not evaluated.’

In the sentence above the proportion of patients should not be in brackets.

‘Children with extrapulmonary tuberculosis were associated with unsuccessful treatment outcomes (AOR 1.64; 95 % CI: 1.02 – 2.62).

The sentence above should start with ‘Extrapulmonary…’.

Introduction

Line 77: Some of the formatting has got lost on the way. I assume this should be a new paragraph?

Line 78: Replace ‘would be’ with ‘is’.

Line 89: ‘…might help in the continuation of making informed decisions…’

Rather ‘will provide evidence to inform decisions’

Methods

Line 88: In this line and lines 98 and 158 you say that you looked at files and in line 100 and somewhere else you say you captured data from the TB registers. Please address this inconsistency. I can’t believe you looked at over 2000 files!

Line 93: You give inclusion and exclusion criteria for the children, but you also need to explain your rationale for your choice of health facilities and address whether this could have introduced some bias.

Line 100: ‘The study employed a total renumeration of all the data..’ This sentence is very confusing and it is unclear exactly what you are trying to say.

Line 161: We try not to use the terminology ‘TB cases’ anymore but rather something like ‘people with TB’ or ‘patients with TB.’

Results

Much of the text in the results section repeats what is in the table. Usually just a couple of significant results from the table are highlighted in the text.

Line 264: Do you need to put a p value if there is also a confidence interval?

Discussion

Line 325: Replace ‘investigations’ with ‘studies’.

Reviewer #2: This is a very interesting and well written manuscript adressing a critical problem of Childhood Tuberculosis Outcomes and Factors Associated with Unsuccessful Treatment Outcomes. I have a few comments:

1. in the abstract: L35 "treattement completed" ,remove comma in between

2.L56 The number of new TB cases were "estimated" and not "anticipated"

3. Can the authors clarify the criteria for selecting the 8 Healthcare facilities

4.L189: add number in ()in addition to % for all results so we can see the numerator/denominator

5.L193: add% at 11.8 for female

6. it might be useful to show the desagregated treatment outcomes per Type of TB( pulmonary /extrapulmonary) , as the main result of the study has shown that extra pulmonary TB is associated with unsuccessful treatment outcome.How many children (%) with extra pulmonary TB died? were Lost to follow up or failed treatment?

7. As 41% of children did not have a TB treatment outcome, how many had pulmonary and extra pulmonary TB? It might be useful to provide the characteristics of this particular group. Are they different compared to those with a treatmemnt outcome?

6. PLOS authors have the option to publish the peer review history of their article (what does this mean?). If published, this will include your full peer review and any attached files.

**Do you want your identity to be public for this peer review?** For information about this choice, including consent withdrawal, please see our Privacy Policy.

Reviewer #1: No

Reviewer #2: **Yes: **Angelique Kany Kany Luabeya

---

## [Decision Letter · Decision Letter 1]

18 Jan 2024

PGPH-D-23-01445R1

Childhood Tuberculosis Outcomes and Factors Associated with Unsuccessful Treatment Outcomes in Selected Public Hospitals of Lusaka Zambia from 2015 to 2019

Dear Dr. Ngosa,

Thank you for submitting your manuscript to PLOS Global Public Health. After careful consideration, we feel that it has merit but does not fully meet PLOS Global Public Health’s publication criteria as it currently stands. Therefore, we invite you to submit a revised version of the manuscript that addresses the points raised during the review process.

Specifically, the reviewer requests further clarification about the bacteriologic testing which included children underwent, request to detail baseline characteristics and reasons for not evaluating treatment outcomes. They also suggest incorporating a flow diagram to illustrate the number of children investigated and other details of the study. 

We look forward to receiving your revised manuscript.

Kind regards,

Jennifer Tucker, PhD

Staff Editor

Journal Requirements:

Additional Editor Comments (if provided):

Reviewers' comments:

Reviewer's Responses to Questions

**Comments to the Author**

1. If the authors have adequately addressed your comments raised in a previous round of review and you feel that this manuscript is now acceptable for publication, you may indicate that here to bypass the “Comments to the Author” section, enter your conflict of interest statement in the “Confidential to Editor” section, and submit your "Accept" recommendation.

Reviewer #2: (No Response)

2. Does this manuscript meet PLOS Global Public Health’s publication criteria? Is the manuscript technically sound, and do the data support the conclusions? The manuscript must describe methodologically and ethically rigorous research with conclusions that are appropriately drawn based on the data presented.

Reviewer #2: Partly

3. Has the statistical analysis been performed appropriately and rigorously?

Reviewer #2: No

4. Have the authors made all data underlying the findings in their manuscript fully available (please refer to the Data Availability Statement at the start of the manuscript PDF file)?

Reviewer #2: No

5. Is the manuscript presented in an intelligible fashion and written in standard English?

Reviewer #2: Yes

6. Review Comments to the Author

Reviewer #2: I have few additional comments

1. Only 11% of the children yielded positive bacteriological results. Establishing and reporting the denominator is crucial, specifically for children who underwent bacteriologic testing. Including children who did not undergo microbiological tests in the denominator would be misleading.

2. A significant proportion of children with extra-pulmonary TB (EPTB) did not undergo bacteriological testing, necessitating clarification of the denominator. Among children with EPTB who underwent bacteriological testing, please confirm those with confirmed bacteriological TB.

3. In the discussion section, the authors omitted to mention other potential reasons for not evaluating treatment outcomes, as recommended by the second reviewer. For instance, if transfers occurred, it might be beneficial to say the reasons behind these transfers, considering that this constitutes a substantial group of children diagnosed with TB who are lost within the healthcare system. As previously suggested, including the baseline characteristics of this group separately would be valuable.

4. Consider incorporating a flow diagram to illustrate the number of children investigated for TB, those excluded, the status of bacteriological tests performed, and other relevant details. This visual aid can enhance the clarity of the text.

7. PLOS authors have the option to publish the peer review history of their article (what does this mean?). If published, this will include your full peer review and any attached files.

**Do you want your identity to be public for this peer review?** For information about this choice, including consent withdrawal, please see our Privacy Policy.

Reviewer #2: No

---

## [Decision Letter · Decision Letter 2]

13 May 2024

PGPH-D-23-01445R2

Childhood Tuberculosis Outcomes and Factors Associated with Unsuccessful Treatment Outcomes in Selected Public Hospitals of Lusaka Zambia from 2015 to 2019

Dear Dr. Ngosa,

Thank you for submitting your manuscript to PLOS Global Public Health. After careful consideration, we feel that it has merit but does not fully meet PLOS Global Public Health’s publication criteria as it currently stands. Therefore, we invite you to submit a revised version of the manuscript that addresses the points raised during the review process.

Please address the suggestions from Reviewer 3 highlighted in their attached PDF comments to improve the clarity of details in the manuscript.

We look forward to receiving your revised manuscript.

Kind regards,

Jennifer Tucker, PhD

Staff Editor

Journal Requirements:

2. Please provide separate figure files in .tif or .eps format only and remove any figures embedded in your manuscript file. Please also ensure all files are under our size limit of 10MB.

Additional Editor Comments (if provided):

Reviewers' comments:

Reviewer's Responses to Questions

**Comments to the Author**

1. If the authors have adequately addressed your comments raised in a previous round of review and you feel that this manuscript is now acceptable for publication, you may indicate that here to bypass the “Comments to the Author” section, enter your conflict of interest statement in the “Confidential to Editor” section, and submit your "Accept" recommendation.

Reviewer #3: (No Response)

2. Does this manuscript meet PLOS Global Public Health’s publication criteria? Is the manuscript technically sound, and do the data support the conclusions? The manuscript must describe methodologically and ethically rigorous research with conclusions that are appropriately drawn based on the data presented.

Reviewer #3: Yes

3. Has the statistical analysis been performed appropriately and rigorously?

Reviewer #3: Yes

4. Have the authors made all data underlying the findings in their manuscript fully available (please refer to the Data Availability Statement at the start of the manuscript PDF file)?

Reviewer #3: Yes

5. Is the manuscript presented in an intelligible fashion and written in standard English?

Reviewer #3: Yes

6. Review Comments to the Author

Reviewer #3: The authors have addressed the earlier comments; however, few minor comments and edits are required, as mentioned in the PDF.

7. PLOS authors have the option to publish the peer review history of their article (what does this mean?). If published, this will include your full peer review and any attached files.

**Do you want your identity to be public for this peer review?** For information about this choice, including consent withdrawal, please see our Privacy Policy.

Reviewer #3: No

---

## [Decision Letter · Decision Letter 3]

19 Jun 2024

PGPH-D-23-01445R3

Childhood Tuberculosis Outcomes and Factors Associated with Unsuccessful Treatment Outcomes in Selected Public Hospitals of Lusaka Zambia from 2015 to 2019

Dear Dr. Ngosa,

Thank you for submitting your manuscript to PLOS Global Public Health. After careful consideration, we feel that it has merit but does not fully meet PLOS Global Public Health’s publication criteria as it currently stands. Therefore, we invite you to submit a revised version of the manuscript that addresses the points raised during the editorial review process.

Though the reviewers comments have been addressed and modifications made to the manuscript, the major limitation of this study due to the high proportion of cases without outcome assessment, cannot be overcome. Suggested changes in the methods, tables and discussion will make the interpretation of results meaningful and therefore necessary for further consideration of this paper for publication.

We look forward to receiving your revised manuscript.

Kind regards,

Sonali Sarkar

Academic Editor

Journal Requirements:

1. Please amend your online Financial Disclosure statement. If you did not receive any funding for this study, please simply state: “The authors received no specific funding for this work.”

2. Please update your online Competing Interests statement. If you have no competing interests to declare, please state: “The authors have declared that no competing interests exist.”

3. "Supplementary Table 1.docx" is/are currently uploaded as an 'Other' file type, which is not viewable by reviewers. Please ensure that all files meant for review are uploaded as 'Supporting Information' and include a legend in the manuscript.

Additional Editor Comments (if provided):

1. Introduction can include

a. the incidence of TB in Zambia, and

b. what proportion of all TB cases in Zambia are in the paediatric age group.

2. Setting description should include

a. The various level hospitals in Zambia with the population covered and what is meant by level 1.

b. Details of the treatment given should be mentioned as there was a shift from alternate-day regimen to daily regimen in many countries during the time of this study.

3. How many were the total cases in the registers? What proportion of all cases had missing information and therefore excluded to reach 2531?

4. A large proportion (40.9%) of all cases (2531) included in the study did not have their outcomes evaluated. This study is about describing the treatment outcomes of paediatric TB and identifying factors associated with it. Therefore the number of cases to be included for this study should be 1495, for whom the treatment outcome is available, and not 2531. The demographic and clinical characteristics should be given for the 1495 cases. Add a column in table 1 for the 1495 who had treatment outcomes. Please modify the text accordingly.

5. Bacteriological confirmation would be relevant for interpreting the treatment outcomes and therefore this description should be provided for the 1495. Description of 2531 cases and the differences in the bacteriological positivity between the groups is not as per the objectives of the study. Replace the denominator for the table 2 with 1495. Please modify the text accordingly.

6. Discussion should include

a. what proportion of the total paediatric TB cases in Zambia were included in this study?

b. to what extent they were representative of all the cases?

c. the differences between 2531 and 1495 in terms of their clinical characteristics.

Reviewers' comments:

Reviewer's Responses to Questions

**Comments to the Author**

1. If the authors have adequately addressed your comments raised in a previous round of review and you feel that this manuscript is now acceptable for publication, you may indicate that here to bypass the “Comments to the Author” section, enter your conflict of interest statement in the “Confidential to Editor” section, and submit your "Accept" recommendation.

Reviewer #3: All comments have been addressed

2. Does this manuscript meet PLOS Global Public Health’s publication criteria? Is the manuscript technically sound, and do the data support the conclusions? The manuscript must describe methodologically and ethically rigorous research with conclusions that are appropriately drawn based on the data presented.

Reviewer #3: Yes

3. Has the statistical analysis been performed appropriately and rigorously?

Reviewer #3: Yes

4. Have the authors made all data underlying the findings in their manuscript fully available (please refer to the Data Availability Statement at the start of the manuscript PDF file)?

Reviewer #3: Yes

5. Is the manuscript presented in an intelligible fashion and written in standard English?

Reviewer #3: Yes

6. Review Comments to the Author

Reviewer #3: (No Response)

7. PLOS authors have the option to publish the peer review history of their article (what does this mean?). If published, this will include your full peer review and any attached files.

**Do you want your identity to be public for this peer review?** For information about this choice, including consent withdrawal, please see our Privacy Policy.

Reviewer #3: No

---

## [Editor Report · Decision Letter 4]

9 Sep 2024

Childhood Tuberculosis Outcomes and Factors Associated with Unsuccessful Treatment Outcomes in Selected Public Hospitals of Lusaka Zambia from 2015 to 2019

PGPH-D-23-01445R4

Dear Mr Ngosa,

We are pleased to inform you that your manuscript 'Childhood Tuberculosis Outcomes and Factors Associated with Unsuccessful Treatment Outcomes in Selected Public Hospitals of Lusaka Zambia from 2015 to 2019' has been provisionally accepted for publication in PLOS Global Public Health.

Best regards,

Sonali Sarkar

Academic Editor